# Cardiac Device-Related Infective Endocarditis Caused by *Salmonella* Infantis—Case Report and Review of Clinical and Epidemiologic Implications

**DOI:** 10.3390/pathogens14050474

**Published:** 2025-05-14

**Authors:** Kristína Doležalová, Lubomír Soják, Annamária Grigláková, Ján Jurenka, Martin Sedlák, Lucia Horniaková, Peter Kromka, Mária Szántová, Peter Sabaka

**Affiliations:** 1Department of Infectology and Geographical Medicine, Faculty of Medicine, Comenius University in Bratislava, Limbova 5, 831 01 Bratislava, Slovakia; dolezalova9@uniba.sk (K.D.); jurenka1@uniba.sk (J.J.); 23rd Department of Internal Medicine, Faculty of Medicine, Comenius University in Bratislava, Limbova 5, 831 01 Bratislava, Slovakia; martinsedlak.ms@gmail.com (M.S.); horniakova22@uniba.sk (L.H.); maria.szantova@kr.unb.sk (M.S.)

**Keywords:** *Salmonella* Infantis, cardiac device-related infective endocarditis, extraintestinal salmonelosis

## Abstract

Background: *Salmonella enterica serovar* Infantis (*S.* Infantis) is a widespread pathogen in agriculture, causing epidemics in chicken flocks. Despite being primarily an animal pathogen, it may pose significant health risks to immunocompromised individuals. Methods: This report describes the first known case of cardiac device-related infective endocarditis (CDRIE) attributed to *S.* Infantis, highlighting its emerging pathogenic potential. It also reviews the literature for microbiologic and epidemiologic perspectives. Results: A 61-year-old male with a history of high-grade multiple myeloma presented with nonspecific symptoms, including low-grade fever and exertional dyspnoea. Blood cultures identified a pure culture of *S.* Infantis, and transoesophageal echocardiography revealed vegetations on pacing leads. Following pacemaker extraction and appropriate antimicrobial therapy, the patient’s condition temporary improved, but later deteriorated due to the progression of underlying malignancy. Conclusions: This case underscores the importance of considering *S.* Infantis in the differential diagnosis of endocarditis in immunocompromised patients, along with the critical need for stringent food safety measures to mitigate infection risks from contaminated poultry products.

## 1. Introduction

Non-typhoidal *Salmonella* is one of the most important foodborne bacterial pathogens [1]. The majority of cases are attributed to the Enteritidis serovar (*S.* Enteritidis). In Europe, *Salmonella enterica* serovar Infantis (*S.* Infantis) is the fourth most common non-typhoidal *Salmonella* serovar responsible for human illness and the most common serovar found in broiler flocks [2,3]. In Slovakia, *S.* Infantis has been detected in several poultry flocks [2,3,4,5,6,7]. Its ability to cause human disease is low, usually only causing illness in severely immunocompromised individuals. Most common clinical presentation is gastroenteritis which is usually self-limiting. More severe infections like osteomyelitis, septic arthritis and abscesses were reported as well [1]. A fundamental feature of *S.* Infantis is its resistance to multiple antibiotics, including third-generation cephalosporins and quinolones. Multidrug-resistant (MDR) strains of *S*. Infantis have been isolated not only from poultry but also from humans [1],. Bacteremia during acute infection (even self-limiting gastroenteritis) can lead to dissemination and colonisation of prosthetic materials such as pacemaker leads [1,8,9]. Cardiac device-related infective endocarditis (CDRIE) is a rare but life-threatening complication of pacemaker implantation [10]. It accounts for approximately 6% of all endocarditis and is associated with significant morbidity and mortality, especially if the device is not removed in time [10,11,12]. The majority of cases are caused by staphylococci or streptococci and gram negative bacteria accounts for less than 6% of CDRIE [11,12,13]. *S.* Enteritidis is a very rare cause of CDRIE and its precise incidence remains unknown [13]. To our knowledge, there have been no reports of CDRIE due to *S.* Infantis.

## 2. Materials and Methods

This report presents the inaugural case of CDRIE associated with *S.* Infantis, thereby elucidating its emerging pathogenic potential. Additionally, it conducts a comprehensive review of the literature from microbiologic and epidemiologic perspectives in order to increase general awareness of this emerging pathogen.

## 3. Results

We present the case of a 61-year-old male (ethnicity: Slovak-West Slavic) with a significant medical history of high-grade IgG kappa myeloma (SLIM-CRAB positive, ISS III, del TP53), who underwent autologous blood marrow transplantation in June 2024. His past medical history also included arterial hypertension, paroxysmal atrial fibrillation, an ascending aortic aneurysm, a bicuspid aortic valve, and sick sinus syndrome.

On 11 December 2024, the patient presented to the Emergency Department of the University Hospital in Bratislava with a clinical history of low-grade fever (up to 37.7 degrees Celsius), progressive weakness, and exertional dyspnoea lasting two months. Prior to admission, he received empirical treatment with ciprofloxacin and amoxicillin-clavulanate prescribed by his general practitioner and haematologist.

Objective examination revealed hypotension (95/65 mm Hg) and a systolic murmur over the aortic valve (Grade 3/6). The remaining physical examination was unremarkable. Laboratory results indicated elevated levels of C-reactive protein (121.1 mg/L), creatinine (111 μmol/L), total bilirubin (22.8 μmol/L), interleukin-6 (26.2 pg/mL), and mild lymphopenia (280 cells/μL). Notably, procalcitonin and conjugated bilirubin levels were unremarkable. Chest radiography identified inhomogeneous opacity in the right lower lobe.

Subsequent computed tomography of the chest revealed a subpleural inflammatory opacity in the lateral aspect of the right lower lobe measuring 130 mm × 115 mm × 22 mm, along with a smaller opacity in the apical region of the right upper lobe measuring 26 mm × 30 mm × 25 mm, with no evidence of cavitation (Figure 1). The patient was admitted with a diagnosis of community-acquired pneumonia and initiated on empirical intravenous antibiotic therapy with cefotaxime (2 g every 8 h). Blood cultures obtained before the initiation of antibiotics subsequently yielded a pure culture of *S.* Infantis (2 sets positive of 3 sets drawn). The pathogen was cultured on deoxycholate citrate agar and finally identified using matrix-assisted laser desorption ionization–time of flight mass spectrometry (MALDI-TOF MS) and slide agglutination serotyping. Antimicrobial susceptibility testing was performed by obtaining Minimal Susceptibility Concentrations (MIC) using antibiotic gradient strips. The isolated *S.* Infantis strain was susceptible to piperacilin-tazobactam, cefotaxime, ceftazidime, cefepime, ertapenem, meropenem, trimetoprim-suphametoxasole and colimycine and resistant to ampicilin, ampicilin-sulbactam, cefuroxime, gentamycin, ciprofloxacin and tetracycline. The MICs of the atibiotcs tested are provided in Table 1.

Given the absence of gastrointestinal symptoms, extraintestinal salmonellosis was suspected. Transthoracic echocardiography (TTE) demonstrated a bicuspid aortic valve and mild mitral and tricuspid regurgitation without signs of vegetations. However, transesophageal echocardiography (TEE) revealed vegetations on the right ventricular pacing leads (8 × 11 mm) and reaffirmed the presence of a bicuspid aortic valve, with otherwise unremarkable findings.

Based on these clinical and echocardiographic findings, a diagnosis of CDRIE was established. Duke criteria for the diagnosis of infective endocarditis were met [14] and there were vegetations visible on pacemaker leads. The patient was subsequently transferred to the National Institute of Heart Diseases in Bratislava, where the pacemaker was explanted on 20 December 2024. Cultures from the electrode revealed a pure culture of *S.* Infantis. Postoperatively, he was returned to the Department of Internal Medicine and later to the Department of Infectious Diseases and Geographical Medicine at the University Hospital in Bratislava, where intravenous cefotaxime treatment was continued. A peripherally inserted central catheter (PICC) was placed for prolonged parenteral antimicrobial therapy.

The patient’s fever resolved, and CRP levels decreased to 44.9 mg/L within ten days following pacemaker extraction. However, on 2 January 2025, the patient experienced a recurrence of fever and productive cough, with an increase in CRP to 144 mg/L. TEE was unremarkable and blood cultures sterile that ruled out the worsening of enodacarditis. Chest radiography detected bilateral pulmonary infiltrates. Thus, a diagnosis of nosocomial pneumonia was established, leading to an escalation of antimicrobial treatment to meropenem (2 g every 8 h). Following this adjustment, the patient exhibited symptomatic relief and a reduction in CRP, prompting de-escalation of therapy to ceftriaxone on 10 January 2025.

On 15 January 2025, the patient presented with chills and fever, reaching a maximum temperature of 39.5 degrees Celsius, accompanied by a significant rise in C-reactive protein (CRP) to 201.5 mg/L. Blood cultures obtained from both peripheral vein and PICC revealed methicillin-resistant *Staphylococcus haemolyticus*, which remained susceptible to tigecycline. In light of these findings, the PICC was removed, and the parenteral tigecycline (50 mg every 12 h) was initiated, while continuing therapy with ceftriaxone. This intervention resulted in the resolution of fever and chills, as well as notable clinical and laboratory improvement.

However, on 19 January 2025, the patient began to report progressive weakness and back pain. Subsequent laboratory evaluations revealed a marked increase in serum creatinine levels to 723 μmol/L and urea levels to 32.6 mg/L, necessitating the initiation of intermittent haemodialysis. Complete blood counts demonstrated severe monocytosis (16,000 cells/μL), as well as the development of significant anemia and thrombocytopenia. Flow cytometric analysis indicated the presence of over 67% atypical plasma cells in the peripheral blood.

Based on these clinical and laboratory findings, a diagnosis of plasma cell leukemia was established. Given the patient’s recent episodes of sepsis and overall frailty, a palliative approach was adopted. The patient ultimately succumbed to his underlying haematological malignancy on 28 February 2025. The duration of antimicrobial therapy targeting *S.* Infantis was 78 days, comprising 68 days of ceftriaxone followed by 10 days of meropenem.

## 4. Discussion

### 4.1. Clinical Consideration

We present the first reported case of CDRIE attributed to *S.* Infantis. CDRIE is a relatively uncommon form of bacterial endocarditis [10] The most common pathogens causing CDRIE are coagulase negative staphylococci, followed by oral streptococci and *Staphylococcus aureus* [11,12]. Although there have been reports of CDRIE linked to *S.* Enteritidis, such occurrences are considered extremely rare and actual incidence is unknown [13]. In comparison to *S.* Enteritidis, *S.* Infantis exhibits a lower virulence. It is predominantly recognized as the important pathogen affecting poultry; however, *S.* Infantis infrequently manifests as disease in humans, predominantly affecting severely immunocompromised individuals [1]. In this case, the patient had a history of high-grade multiple myeloma and had undergone chemotherapy and autologous blood marrow transplantation, categorizing him as severely immunocompromised. The patient initially presented with nonspecific extraintestinal symptoms, including low-grade fever, progressive weakness, and exertional dyspnoea. The diagnosis of CDRIE was established based on positive blood cultures and echocardiographic findings of vegetations on the pacemaker leads identified via TEE, following negative transthoracic echocardiography TTE. This case underscores the necessity for heightened vigilance and awareness regarding CDRIE caused by opportunistic pathogens in the differential diagnosis of fever of unknown origin in immunocompromised patients with implanted pacemakers.

### 4.2. Diagnosis and Management of CDRIE

CDRIE is a relatively rare complication of permanent transvenous pacing. It accounts for approximately 6% of all endocarditis [10,12,15]. It is a severe and potentially life-threatening condition. The reported lethality of CDRIE varies from 13% to 66%. The fatal outcome is much more likely to occur in cases where the pacemaker leads are left in place [10,12,15,16].

The clinical presentation of CDRIE is variable and the diagnosis can be elusive. The most common presentation is fever, present in more than 86% of patients. Blood cultures are usually positive, especially in patients not receiving antimicrobial therapy. The diagnosis is usually made on the basis of positive blood cultures and visualisation of vegetations on pacemaker leads [10,15]. The presence of at least 2 separate blood cultures and evidence of endocardial involvement represent major Duke criteria for infective endocarditis [14,17]. On the other hand, several minor Duke criteria may be absent. As the vegetations are located in the right ventricle, peripheral arterial emboli are usually absent, but septic pulmonary emboli or pneumonia are present, and immunogenic phenomena such as Osler’s nodules and Roth spots may occur [18]. The audible murmur is rare, the clinical presentation may be non-specific, and visualisation of vegetations is essential for diagnosis. The yield of transthoracic echocardiography for vegetations in CDRIE has been shown to be as low as 22%. On the other hand, the sensitivity of TEE exceeds 90% [19]. Therefore, in patients with possible CDRIE, TEE should be performed if TTE is negative. Blood cultures are also essential for definitive diagnosis. As mentioned above, 2 separate positive blood cultures are one of the main criteria for endocarditis. They are also essential for the identification of the etiopathogen, which facilitates the tailoring of appropriate antimicrobial treatment [18].

The management of CDRIE necessitates the removal of the cardiac device system, in conjunction with initial empiric antibiotic therapy aimed at targeting *Staphylococcus* species and Gram-negative bacteria until the specific pathogen is identified [20]. While there is ongoing debate regarding the necessity of lead extraction in all cases, evidence indicates a higher mortality rate in patients who do not undergo lead extraction [12,14,15,16]. Consequently, the complete removal of all components of the pacemaker is strongly recommended [20].

The antimicrobial treatment regimens and their duration do not differ significantly from those used for other forms of infective endocarditis. Empiric antimicrobial therapy is administered initially and is subsequently followed by pathogen-directed treatment once the results of blood cultures are available. Typically, a duration of two to four weeks of antimicrobial therapy is considered sufficient; however, in instances where complete extraction of all components of the cardiac device is not feasible, the antimicrobial treatment may be extended to six weeks or longer [10,20].

### 4.3. Microbiological Perspective

Worldwide, non-typhoidal *Salmonella* is responsible for an estimated 94 million infections and more than 200,000 deaths per year, making it one of the most important foodborne bacterial pathogens [8,21,22]. *Salmonella* is a genus of Gram-negative motile bacilli belonging to the family *Enterobacteriaceae*. These bacilli do not ferment lactose, use citrate as a sole carbon source and lysine as a nitrogen source, and produce hydrogen sulphide, which can be detected on Triple Sugar-Iron (TSI) agar (black coloration of the lower part of the medium) or on deoxycholate citrate agar, on which they grow as lactose-negative, black-colored colonies [23].

The genus *Salmonella* consists of two species, *Salmonella* bongori and *S.* enterica [24]. The species *Salmonella* enterica consists of six subspecies, enterica, salamae, arizonae, diarizonae, houtenae, and indica, with approximately 2659 serovars. Of these, the enterica subspecies represents approximately 1547 serovars, of which 99% can cause infections in both humans and animals [25]. *Salmonella* causes mainly gastroenteritis, but it can also spread outside the gastrointestinal tract as the acute infection may be associated with bacteraemia. Extraintestinal salmonellosis can affect various organ systems and cause osteomyelitis, meningitis, aortitis, mycotic aneurysm and sepsis. The vast majority of cases of salmonellosis are caused by the serovar Enteritidis. Other serotypes such as *S.* Infantis are rare [23].

*S.* Infantis is the fourth most common non-typhoidal *Salmonella* causing human disease in Europe [3]. In Europe, *S.* Infantis has been reported as the most common *Salmonella* serovar in broiler flocks (45.6%) and broiler meat (50.6%) [2,3]. In Slovakia, it has been detected in several poultry flocks in 2017 [3,4,6]. Its pathogenic potential compared to *S.* Enetritidis in the human host is believed to be minimal. It usually causes disease only in a severely immunocompromised host [1]. The epidemiological significance of *S.* Infantis is predominantly attributed to its resistance to multiple antibiotics, including critically important third-generation cephalosporins, aminoglycosides and quinolones, which may severely restrict therapeutic options [7]. *S.* Infantis is characterized by the presence of a megaplasmid (pESI), which often harbours genes encoding for extended-spectrum beta-lactamases (ESBL), such as bla_CTX-M-65_ [26]. These broad-spectrum beta-lactamases effectively inactivate third-generation cephalosporins, complicating the management of infections caused by this pathogen [27]. The strain of *S.* Infantis isolated from our patient was resistant to aminoglycosides, ciprofloxacin, and ampicillin-sulbactam. However, it was susceptible to third-generation cephalosporins and was not the ESBL producer. Penicillins, aminoglycosides, and vancomycin are often used as first-line treatments for cardiac device-related infective endocarditis [20]. Given the resistance profile of *S.* Infantis, these antibiotics may be ineffective in treating infections caused by this pathogen. Therefore, caution should be exercised when using them in severely immunocompromised adults. The occurrence of *S.* Infantis as a human pathogen is concerning, given its widespread distribution and resistance to both first- and second-line treatment alternatives. This extensive antibiotic resistance profile positions MDR *S.* Infantis as a rising public health threat [7].

### 4.4. Epidemiology of S. Infantis

Worldwide, serovar *S.* Infantis is one of the most common serovars isolated from both food and animal sources, especially broilers [2,3,4,5,28,29,30,31]. Given these facts, in recent years serovar *S.* Infantis has also become a relevant agent of salmonellosis in humans [3,32,33,34]. Large outbreaks have been associated with food products such as raw or undercooked chicken, pork, beef and eggs [35,36,37,38,39]. These strains are able to circulate for periods of more than 10 years, indicating high clonal stability [40,41]. *S.* Infantis has the ability to tolerate adverse physical conditions (for example dehydration) relatively well, which may help to increase its ability to survive in the food chain [40]. *S.* Infantis has been isolated from contaminated livestock feed and pet food (e.g., dog chews and dry dog food, beef trimmings, or oilseeds) [42]. *S.* Infantis infections in chickens, pigs, and cattle are typically asymptomatic and are primarily detected through laboratory testing. Additionally, asymptomatic food handlers have been associated with human infections and outbreaks, resulting from the transmission of contaminated food products [43]. *S.* Infantis can be transmitted not only indirectly through food, but also directly from person to person and from infected animal to person [42,44].

In 2023, official control authorities in Slovak Republic examined more than 11,000 food samples, revealing a positivity rate of 0.54%, with higher percentages found in unspecified poultry meat (13%), broiler meat (6.12%), and eggs (5.38%). Among the 544 poultry meat samples, 6.12% of 376 broiler samples and 13% of 131 unspecified poultry meat samples tested positive for *Salmonella* spp. Targeted inspections focused on meat in catering establishments identified several *Salmonella* serovars including *S.* Infantis [6]. Also, more than 8000 clinical animal samples were analyzed, with 1.3% of 6331 poultry flocks testing positive, predominantly for *S.* Infantis. In companion animals, 1.7% of dog samples and 1.9% of cat samples also showed positivity for *S.* Infantis, likely due to the consumption of insufficiently cooked poultry meat. Additionally, 863 feed samples were assessed, with 4 positives identified for *Salmonella* spp., including *S.* Infantis [6]. An 11-month-old female patient was infected with *S.* Infantis after exposure to a lizard’s litter [6]. Lithuania reported on 5 July 2024 the presence of *S.* Infantis in frozen chicken inner fillets produced in Slovakia, from chickens slaughtered in Ukraine [4]. The Czech Republic reported in 2024 the presence of *S.* Infantis in frozen chicken breasts, which were originally shipped from Poland via the Slovak Republic [5,45].

Current evidence indicates that *S.* Infantis may be present in commercially available chicken meat and meat products in Europe, posing a significant health threat to severely immunocompromised patients. This underscores the importance of exercising caution in the preparation and consumption of meat and meat products, particularly poultry, by individuals with compromised immune systems.

### 4.5. Strengths and Limitations

The primary strength of our case report lies in its originality, as it describes the first documented case of CDRIE caused by *S.* Infantis. Another notable strength is the inclusion of a review of epidemiological studies, which provides a solid contextual background for understanding the significance of this case. A limitation of our study is the absence of sequencing or other genetic analyses of the isolated *Salmonella* strain. Such analyses could have offered deeper insight into the epidemiological origin of the pathogen and supported its precise identification as *S*. Infantis. We were also unable to obtain the patient’s perspective on the treatment he received, as he died during the course of hospitalization.

## 5. Conclusions

*Salmonella enterica* subsp. *enterica* serotype Infantis, along with other serotypes primarily associated with poultry, can act as opportunistic pathogens capable of causing severe infections in immunocompromised individuals. We report a case of CDRIE caused by *S*. Infantis, highlighting its potential role as an etiological agent in CDRIE. The identification of this organism in blood cultures necessitates subsequent echocardiographic assessment, with TEE recommended if TTE yields negative results despite clinical suspicion. Furthermore, given that non-typhoidal *Salmonella* serotypes, including *S.* Infantis, can originate from contaminated poultry, stringent food safety measures should be emphasized in the preparation of meat and meat products for immunocompromised individuals, to mitigate the risk of infection.

## Figures and Tables

**Figure 1 pathogens-14-00474-f001:**
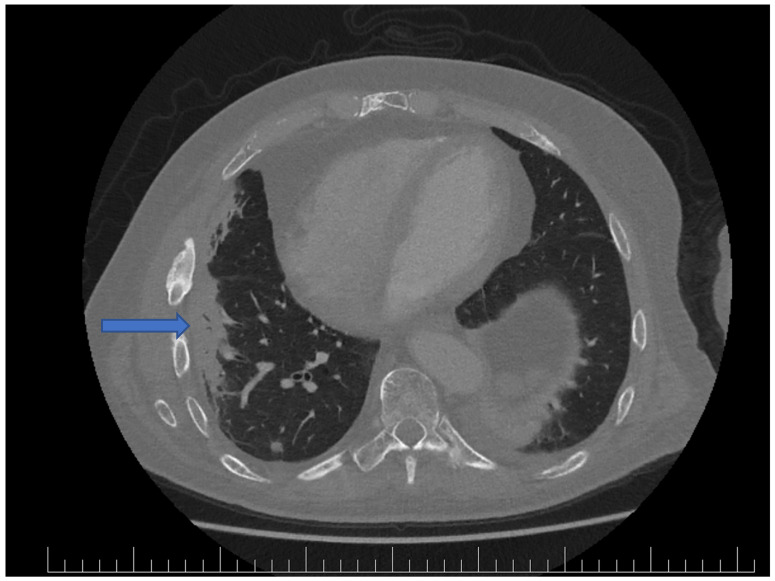
Computed tomography of the chest. It revealed a subpleural inflammatory opacity in the lateral aspect of the right lower lobe measuring 130 mm × 115 mm × 22 mm with no evidence of cavitation (blue arrow).

**Table 1 pathogens-14-00474-t001:** Results of susceptibility testing of *S.* Infantis cultivated from the blood culture. Minimal inhibition concentrations (MIC) and susceptibility were interpreted according to Clinical and Laboratory Standards Institute standards [3].

Susceptible	MIC (mg/L)	Cut-Off Susceptible	Cut-Off Resistant	Resistant	MIC (mg/L)	Cut-Off	Cut-Off Resistant
Piperacilin-tazobactam	2	≤8	≥32	Ampicilin	>32	≤8	>32
Cefotaxime	0.25	≤1	≥4	Ampicilin-sulbactam	>16	≤8	≥32
Ceftazidime	0.5	≤4	≥16	Cefuroxime	>32	≤8	≥32
Cefepime	0.25	≤2	≥16	Gentamycin	>32	≤4	≥16
Ertapenem	0.03	≤4	≥16	Tetracyclin	>32	≤4	≥16
Meropenem	0.12	≤4	≥16	Ciprofloxacin	2	≤0.06	≥2
Trimetoprim-sulphametoxasole	1	≤2	≥4				
Colimycine	0.05	≤2	≥4				

## Data Availability

The data can be made available on reasonable request.

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
