# Peer review of "Cardiac Device-Related Infective Endocarditis Caused by *Salmonella* Infantis—Case Report and Review of Clinical and Epidemiologic Implications"

_pathogens, 2025, doi:10.3390/pathogens14050474_

Round 1

Reviewer 1 Report

Comments and Suggestions for Authors

The Care Report by Dolezalova, K. et al., is an interesting report on an Infective Endocarditis case caused by Salmonella Infantis. The authors have done an excellent job in presenting the case and reviewing the appropriate literature. However, I have the following minor comments:

  1. Please review the nomenclature used to describe Salmonella species so that it is consistent with international bacterial nomenclature. For example, on Line 12, you should use the following: Salmonella enterica subspecies enterica serovar Infantis (S. Infantis) the first time followed by S. Infantis for all subsequent references (note Infantis is not italicized). Please review the entire case report to ensure that it is consistent. Additionally there are a few instances where a bacterial name is mentioned but not italicized like on Line 188 where Staphylococcus is not italicized. On Line 215, Salmonella is mentioned but not italicized. Please review the entire case report to ensure consistency.
  2. On L79, "citrate agarand", I think this needs to be checked as there is a space missing here. As well on the same line "finaly" and "indetified" are typos that should be corrected. Please proofread the entire case report to ensure there are no typos.
  3. For Reference 8, the author list is a page and a half long. This might be something to address with the journal editors and/or the journal reference requirements.

Author Response

Dear reviewer,

We sincerely thank you for your thorough review and thoughtful comments on our manuscript. We appreciate the time and effort you invested in evaluating our work and the valuable feedback you provided.

We have carefully considered each of your suggestions and have revised the manuscript accordingly. Below, we provide a detailed, point-by-point response to each of your comments, indicating how we have addressed them in the revised version of the manuscript.

Comments:  The Care Report by Dolezalova, K. et al., is an interesting report on an Infective Endocarditis case caused by Salmonella Infantis. The authors have done an excellent job in presenting the case and reviewing the appropriate literature. However, I have the following minor comments:

Please review the nomenclature used to describe Salmonella species so that it is consistent with international bacterial nomenclature. For example, on Line 12, you should use the following: Salmonella enterica subspecies enterica serovar Infantis (S. Infantis) the first time followed by S. Infantis for all subsequent references (note Infantis is not italicized). Please review the entire case report to ensure that it is consistent. Additionally there are a few instances where a bacterial name is mentioned but not italicized like on Line 188 where Staphylococcus is not italicized. On Line 215, Salmonella is mentioned but not italicized. Please review the entire case report to ensure consistency.

Response: We corrected the nomenclature in the manuscript according to reviewer comments.

On L79, "citrate agarand", I think this needs to be checked as there is a space missing here. As well on the same line "finaly" and "indetified" are typos that should be corrected. Please proofread the entire case report to ensure there are no typos.

Resposne: We corrected the typos in line 79. Incorect word „agarnad“ was replaced by „agar and“ and corrected other typos in the manuscript like "finaly" in line 79.

For Reference 8, the author list is a page and a half long. This might be something to address with the journal editors and/or the journal reference requirements.

Response: We tried to use the citing style recomneded by the journal submission instructions and used Zotero for automation. However, unfortunately we missed the recomendation to shorten the author lists after first ten authors. We correceted the refference list accotding to recomendation.  

We hope that the changes we have made meet your expectations and improve the quality and clarity of the manuscript. Thank you again for your constructive input, which has been instrumental in refining our work.

Sincerely,

Peter Sabaka

On behalf of all co-authors

Reviewer 2 Report

Comments and Suggestions for Authors

Major points

- line 36 “Its ability to cause human disease is low, usually only causing illness in severely immunocompromised individuals” – also add the infections caused by this pathogen in immunocompromised patients to provide a better overview

- line 37 “A fundamental feature of S. infantis is its resistance to multiple antibiotics, including third-generation cephalosporins and quinolones” – multiple other studies also highlighted the increased spread of MDR serovars in both humans and poultry (e.g., 10.3201/eid3004.231031)

- line 45 – „S. enteritidis is a very rare cause of CDRIE” – expand this statement; how rare? In addition, how frequent are CDRIE cases determined by other Salmonella serovars?

- line 77 – please include essential details regarding the blood cultures, such as the number of sets drawn, the percentage that yielded positive results, and the time to positivity.

- line 80 - Also add the confidence score provided by MALDI-TOF MS for the identification

- line 80 – was the identification not confirmed through serotyping or other methods? As far as I am aware of, MALDI-TOF MS is reliable only in the case of species level identification in Salmonella, but not for typing/subtyping (e.g., 10.1177/1469066717699216); In my opinion, this represents a significant limitation of the study; if the isolate is still available (e.g., frozen at -80C), I would recommend confirming the identification through another method

- line 82 – add the standard used for MIC interpretation and the respective cut-off values in Table 1

- line 100 – „cefotaxime treatment was resumed” – the timeline is not entirely clear; was the treatment with cefotaxime stopped at any point and resumed afterwards? What was the overall duration of the treatment regimen?

- line 233 – add all resistances here

- Discussion – if the Ministry of Agriculture and Rural Development of the Slovak Republic, 2024 was added as a reference, there is no need to also specify it in the text multiple times

- Conclusions – even if the identification would have been confirmed through another method, I do not believe that one case qualifies S. Infantis as “a potential pathogen” in the case of CDRIE

- CARE checklist 11a – strengths and limitations seem to be missing

- CARE checklist 12 – similar as above; patient perspective on the treatment is marked as available but missing

Minor points

- Table 1 – minor editing required (ertapenem is overlapped with a number)

- line 240 “The emergence of multidrug-resistant (MDR) The occurence of S. infantis as a human”?

- some references require formatting

Author Response

Dear reviewer,

We sincerely thank you for your thorough review and thoughtful comments on our manuscript. We appreciate the time and effort you invested in evaluating our work and the valuable feedback you provided.

We have carefully considered each of your suggestions and have revised the manuscript accordingly. Below, we provide a detailed, point-by-point response to each of your comments, indicating how we have addressed them in the revised version of the manuscript.

Comments and revisions:

Major points

- line 36 “Its ability to cause human disease is low, usually only causing illness in severely immunocompromised individuals” – also add the infections caused by this pathogen in immunocompromised patients to provide a better overview

Revision: we addede most common types of reported infections caused by S. Infantis in immunocompromised hosts.

- line 37 “A fundamental feature of S. infantis is its resistance to multiple antibiotics, including third-generation cephalosporins and quinolones” – multiple other studies also highlighted the increased spread of MDR serovars in both humans and poultry (e.g., 10.3201/eid3004.231031)

Revision: We added that the MDR strains were isolated not only in poulty but also in humans and added the citation you recommanded.

- line 45 – „S. enteritidis is a very rare cause of CDRIE” – expand this statement; how rare? In addition, how frequent are CDRIE cases determined by other Salmonella serovars?

Revisions: The garm-negative bacteria are causing less than 6% of CDRIE and exact percentage of CRIE caused by Salmonella spp. Is undetermined. We added to the text: „The majority of cases are caused by staphylococci or streptococci and gram negative bacteria accounts for less than 6% of CDRIE 11-13. S. Enteritidis is a very rare cause of CDRIE and its precise incidence remains unknown. 13. To our knowledge, there have been no reports of CDRIE due to S. Infantis.“

- line 77 – please include essential details regarding the blood cultures, such as the number of sets drawn, the percentage that yielded positive results, and the time to positivity.

Revisions: 2 of3  sets were positive for S. Infantis. We added this information to the text.

- line 80 - Also add the confidence score provided by MALDI-TOF MS for the identification

Revisions: Unfortunately our comertional laboratory does not provide us with confidence scores. However, we added the information that S. Infantis were identified alsio using Serotyping.

- line 80 – was the identification not confirmed through serotyping or other methods? As far as I am aware of, MALDI-TOF MS is reliable only in the case of species level identification in Salmonella, but not for typing/subtyping (e.g., 10.1177/1469066717699216); In my opinion, this represents a significant limitation of the study; if the isolate is still available (e.g., frozen at -80C), I would recommend confirming the identification through another method

Revisions: we added the information that S. Infantis were identified alsio using slide agglutination serotyping.

- line 82 – add the standard used for MIC interpretation and the respective cut-off values in Table 1

Revisions: We added standards (Clinical and Laboratory Standards Institute privided by ECDC) and cut-offs in to the table 1.

- line 100 – „cefotaxime treatment was resumed” – the timeline is not entirely clear; was the treatment with cefotaxime stopped at any point and resumed afterwards? What was the overall duration of the treatment regimen?

- line 233 – add all resistances here

Revisions: we added Cefuroxime which missed from the sentence.

- Discussion – if the Ministry of Agriculture and Rural Development of the Slovak Republic, 2024 was added as a reference, there is no need to also specify it in the text multiple times

Revisions: It was a mistake done by the Zotero. We revised the text accoridng to your recomendation.

- Conclusions – even if the identification would have been confirmed through another method, I do not believe that one case qualifies S. Infantis as “a potential pathogen” in the case of CDRIE

Revisions: We made our conclusion much less definitive. Adjusted claim: „Salmonella enterica subsp. enterica serotype Infantis, along with other serotypes primarily associated with poultry, can act as opportunistic pathogens capable of causing severe infections in immunocompromised individuals. We report a case of CDRIE caused by S. Infantis, highlighting its potential role as an etiological agent in CDRIE.“

- CARE checklist 11a – strengths and limitations seem to be missing

Revision: We added caption Strenghts and limitations: The primary strength of our case report lies in its originality, as it describes the first documented case of CDRIE caused by S. Infantis. Another notable strength is the inclusion of a review of epidemiological studies, which provides a solid contextual background for understanding the significance of this case. A limitation of our study is the absence of sequencing or other genetic analyses of the isolated Salmonella strain. Such analyses could have offered deeper insight into the epidemiological origin of the pathogen and supported its precise identification as S. Infantis.

- CARE checklist 12 – similar as above; patient perspective on the treatment is marked as available but missing

Revisions: In limations, we added statement: We were also unable to obtain the patient’s perspective on the treatment he received, as he died during the course of hospitalization.

Minor points

- Table 1 – minor editing required (ertapenem is overlapped with a number)

Revisions: We performed thorough stylisation and grammar checking.

- line 240 “The emergence of multidrug-resistant (MDR) The occurence of S. infantis as a human”?

Revisions: We corrected the sentence.

- some references require formatting

Revisions: We formated the reference according to journal recomndations

We hope that the changes we have made meet your expectations and improve the quality and clarity of the manuscript. Thank you again for your constructive input, which has been instrumental in refining our work.

Sincerely,

Peter Sabaka

On behalf of all co-authors

Round 2

Reviewer 2 Report

Comments and Suggestions for Authors

The manuscript has been improved during this round of revisions.

However, one aspect regarding the cefotaxime treatment seems to have been omitted.

line 100 – „cefotaxime treatment was resumed” – the timeline is not entirely clear; was the treatment with cefotaxime stopped at any point and resumed afterwards? What was the overall duration of the treatment regimen?

Author Response

Dear reviewer, 

We sincerely thank you for your thorough review and thoughtful comments on our manuscript. We appreciate the time and effort you invested in evaluating our work and the valuable feedback you provided. 

line 100 – „cefotaxime treatment was resumed” – the timeline is not entirely clear; was the treatment with cefotaxime stopped at any point and resumed afterwards? What was the overall duration of the treatment regimen?

Revisions: Thank you very much that you uncover this discrepancy that we missed. The statement that the therapy was "resumed" is incorrect and was mistakenly made during translation and stylistic polishing. The therapy was "continued", which we corrected in the manuscript. We added the information about the duration of antimicrobial treatment. W added: The duration of antimicrobial therapy targeting Salmonella Infantis was 78 days, comprising 68 days of ceftriaxone followed by 10 days of meropenem.

We hope that the changes we have made meet your expectations and improve the quality and clarity of the manuscript. Thank you again for your constructive input, which has been instrumental in refining our work.

Sincerely,

Peter Sabaka

On behalf of all co-authors